# Lamellar Polypyrene Based on Attapulgite–Sulfur Composite for Lithium–Sulfur Battery

**DOI:** 10.3390/membranes11070483

**Published:** 2021-06-29

**Authors:** Jing Wang, Riwei Xu, Chengzhong Wang, Jinping Xiong

**Affiliations:** 1Beijing Key Laboratory of Electrochemical Process and Technology for Materials, Beijing University of Chemical Technology, Beijing 100029, China; xurw@mail.buct.edu.cn (R.X.); czwang@mail.buct.edu.cn (C.W.); 2Fujian Provincial Key Laboratory of Ecological Industry Green Technology, Wuyi University, Wuyishan 354399, China

**Keywords:** lithium–sulfur batteries, lamellar polypyrene, attapulgite, cathode materials

## Abstract

We report on the preparation and characterization of a novel lamellar polypyrrole using an attapulgite–sulfur composite as a hard template. Pretreated attapulgite was utilized as the carrier of elemental sulfur and the attapulgite–sulfur–polypyrrole (AT @400 °C–S–PPy) composite with 50 wt.% sulfur was obtained. The structure and morphology of the composite were characterized with infrared spectroscopy (IR), thermogravimetric analysis (TGA), and scanning electron microscopy (SEM). An AT @400 °C–S–PPy composite was further utilized as the cathode material for lithium–sulfur batteries. The first discharge specific capacity of this kind of battery reached 1175 mAh/g at a 0.1 C current rate and remained at 518 mAh/g after 100 cycles with capacity retention close to 44%. In the rate test, compared with the polypyrrole–sulfur (PPy–S) cathode material, the AT @400 °C–S–PPy cathode material showed lower capacity at a high current density, but it showed higher capacity when the current came back to a low current density, which was attributed to the “recycling” of pores and channels of attapulgite. Therefore, the lamellar composite with special pore structure has great value in improving the performance of lithium–sulfur batteries.

## 1. Introduction

The theoretical specific capacity of a lithium–sulfur (Li–S) battery is about 1672 mAh/g [1], the determination of which has been the focus of research in recent years. However, such batteries also have several problems. For example, the intermediates are easy to dissolve in the organic electrolyte, causing the “shuttle effect”, which weakens the cycle stability of the battery [2,3,4].

To solve this problem, researchers chose materials with a higher specific surface area and larger pore volume to be the cathode material, so that the more active sulfur could be absorbed and the intermediates during the charge–discharge process could be prevented from dissolving in the electrolyte [5]. Constructing a 3D nanostructure has always been a popular method for the optimization of the cathode material [6]. Yao et al. [7] proposed to use a materials genome database to analyze, guide, and design the composite, which could promote battery efficiency and cost savings. Barchasz et al. [8] adopted mesoporous silica to absorb sulfur and effectively inhibit the dissolution of polysulfides. However, the cost of this nano-oxide was high and the process was complex. Recently, researchers have tried to combine polymers with inorganic materials to form the cathode materials in a Li–S battery, which would have both the advantages of the two materials and lower the cost [9].

Polypyrene is a kind of common conductive polymer (CP). The electronic conjugation between each repeat unit in CPs creates semiconducting molecular wire architectures that impart CPs with interesting optical and electronic properties [10]. Polypyrene has high conductivity and environmental stability and can be prepared into nanomaterials with special properties and different morphologies by designing and selecting different templates [11,12]. Attapulgite (AT) is a kind of hydrated magnesia alumina silicate mineral with the molecular formula of Mg_5_Si_8_O_20_(OH)_2_(OH_2_)_4_·4H_2_O (molecular structure shown as Figure 1). Thanks to its special lamellar and porous structure, which includes what are known as “natural nanotubes”, it has a strong adsorption capacity. It not only has regular channels and pores, but also contains active silicon and aluminum hydroxy [13,14,15,16] on its surface. Thus, it can be widely used in functionalized polymer materials.

In this paper, raw AT soil was purified and modified to be the carrier of elemental sulfur. In addition, polypyrrole (PPy) was prepared on the template of the modified attapulgite–sulfur (AT @400 °C–S). The sulfur content of the composites reached up to 50 wt.%. When the modified attapulgite–sulfur–polypyrrole (AT @400 °C–S–PPy) composite was used as the cathode material, the specific discharge capacity of the battery reached 1175 mAh/g in the first cycle, and it remained above 518 mAh/g after 100 cycles, with a capacity retention rate of 44%. Compared to the contrasting polypyrrole–sulfur (PPy–S) cathode material, the specific capacity of this cathode material was lower at a high rate but higher at a low rate, which definitely showed that the “recycling” of pores and channels of AT can effectively prohibit the “shuttle effect” and improve the performance of Li–S batteries.

## 2. Experiment

### 2.1. Material Preparation

Firstly, untreated raw AT soil (120 mesh) was crushed, milled, and sieved for use. A proper amount of dispersant sodium hexametaphosphate (SHMP, AR, Changsha Tanghua Chemical Trade Co., Ltd., Changsha, China) was added into 200 mL deionized water; then, the solution was stirred, and 20 g AT was slowly added. After that, the mixture was ultrasonically dispersed for 30 min and kept still for 1 h. The upper liquid was decanted to obtain solid precipitation, which was washed and dried at 80 °C in a vacuum drying oven (DZF 2001 type, Shanghai Yiheng Instrument Co., Ltd., Shenyang, China) to obtain the purified AT. The purified AT was milled again and was added into a 150 mL dilute hydrochloric acid solution (HCl, AR, Qingzhou Shengyang Chemical Co., Ltd., Shenyang, China). The mixture was stirred for 60 min and kept still for 30 min to stratify. The upper liquid was decanted to get solid precipitation, which was washed and dried. The dried AT was divided into three parts that were separately heated at 200, 400, and 600 °C for 24 h to obtain the modified ATs.

Then, the untreated raw AT (AT^1^), purified AT (AT^2^), and modified AT (AT heated at 200 °C: AT @200 °C; AT heated at 400 °C: AT @400 °C; AT heated at 600 °C: AT @600 °C) were respectively ground with sulfur (S, Beijing Yili Fine Chemicals Co., Ltd., Beijing, China), evenly mixed, and heated at 105 °C in the tube furnace (JGF1200-120, Shanghai, China). Flowing nitrogen was continuously blown into the tube furnace for 3 h. Five kinds of sulfur-containing AT (sulfur-containing AT^1^: AT^1^–S; sulfur-containing AT^2^: AT^2^–S; sulfur-containing AT @200 °C: AT @200 °C–S; sulfur-containing AT @400 °C: AT @400 °C–S; sulfur-containing AT @600 °C: AT @600 °C–S) were obtained and weighed. The sulfur adsorption amount can be calculated according to the mass difference of the mixture: 1−(W_1_ (the mass of the mixture before absorption)−W_2_ (the mass of the mixture after absorption))/W_2_.

After material optimization (Section 3.1), AT @400 °C–S was chosen to be the template to prepare PPy. Pyrrole monomer (99%, Shanghai Kefeng Chemical Reagent Co., Ltd., Shanghai, China) was dissolved in 100 mL ethanol aqueous solution (AR, Beijing chemical plant), and stirred at 0 °C until it was completely dissolved. The milled AT @400 °C–S was added and stirred for 2 h, and then 0.01 mol ammonium persulfate (APS, AR, Beijing Yili Fine Chemicals Co., Ltd., Beijing, China) solution was added slowly. After 24 h, the products were washed and extracted repeatedly to obtain the black powder. The black powder was dried at 50 °C for 24 h to obtain the final product: AT @400 °C–S–polypyrrole (AT @400 °C–S–PPy).

A contrast sample of PPy–S was also prepared. Oxidant APS was added into the pyrrole–ethanol water solution to synthesize PPy. Sulfur was heated with PPy to prepare PPy–S, which was ready for the electrochemical comparison.

### 2.2. Characterization

The AT @400 °C–S–PPy composite was further characterized by Fourier-transform infrared spectroscopy (FTIR, Nicolet-60 SXB, Thermo Nicolet, Waltham, MA, USA) to determine the elemental composition. The product was subjected to thermogravimetric analysis (TGA, TA-Q50, New Castle, DE, USA) at a heating rate of 10 °C/min to confirm the sulfur content. Scanning electron microscopy (SEM, Zooma 200, Eindhoven, The Netherlands) was used to observe the surface topography. The BET nitrogen adsorption method was used to determine the pore size distribution with the carrier-gas flow of 34.52 mL/min.

### 2.3. Electrochemical Measurement

The AT @400 °C–S–PPy composite and contrast sample PPy–S were separately mixed with conductive agent (conductive graphite, Tianyuan graphite Co., Ltd., Qingdao, China) and binder (polyvinylidene fluoride, PVDF, SOLVAY-SOLEF-9007, Qingdao, China) at the mass ratio of 8:1:1. Then, a small amount of deionized water and ethanol absolute were added. The mixture was water-bath heated for 8 h to obtain uniform viscous slurry. The slurry was uniformly applied to an aluminum foil current collector with an applicator blade. After being dried at 60 °C for 24 h, it was punched into circular electrode pieces at a diameter of about 1 cm. With these punched pieces used as a positive electrode, the lithium piece (16 × 0.6 mm, Taizhou Welding Materials, Taizhou, China) used as a negative electrode, the Celgard 2400 microporous film used as a separator, and 1 mol/L lithium bis((trifluoromethyl) sulfonyl) azanide (LiTFSI; 1,3-dioxolane, DOL/dimethoxyethane, DME, volume ratio of 1:1) used as an electrolyte, button batteries were assembled in the glove box (Lab 2000). The assembled batteries were electrochemically tested after standing still overnight. Constant current charge and discharge testing was carried out in the LAND CT 2001 A Blue Test System (Wuhan, China). The specific capacity and current multiplying ratio of this composite positive electrode were calculated for the mass of the active substance sulfur.

## 3. Results and Discussion

### 3.1. Optimum Conditions for Sulfur Adsorption of AT

The sulfur adsorption amount in the AT^1^–S, AT^2^–S, AT @200 °C–S, AT @400 °C–S, and AT @600 °C–S systems was measured. The results are shown in Table 1 below:

From the data, it is obvious that the sulfur absorption amount of AT increased after AT was purified. According to DLVO theory [17,18], the dispersant SHMP should greatly enhance the repulsion between the AT particles [19], making the AT effectively dispersed. (DLVO theory is named after Derjaguin, Landau, Verwey, and Overbeek. It is an explanation of the stability of colloidal suspension and describes the balance between two forces: electrostatic repulsion and van der Waals attraction.) Thus, the impurities like SiO_2_ in AT can be removed easily and the interlayer force of AT can be weakened, which is conducive to the diffusion of sulfur, improving the adsorption performance. The sulfur absorption amount increased after purified AT was treated through acidification and heat. The impurities hidden in the pores and channels of AT can be removed through acidification. Due to the cation exchange, H^+^ replaced the cations of K^+^, Na^+^, Ca^2+^, and Mg^2+^ in the interlayers of AT. As the radius of H^+^ was smaller, the pore volume of AT became larger [20]. Heat treatment can remove different kinds of water (like absorbed water, zeolite water, crystal water, and structure water) from AT, making its internal structure more porous. Therefore, the specific surface area became larger and the adsorption capacity enhanced.

From Table 1, we can also see that the sulfur absorption amount of AT increased first and then decreased with the increase in temperature. The largest sulfur absorption amount of AT was about 58%, under the conditions that AT had been purified by dispersant SHMP, acidified, and then heated at 400 °C, since, below 100 °C, absorbed water on the AT surface was removed, which would not change the structure and morphology of AT too much. Between 160–230 °C, zeolite water was removed from AT, causing the channels to be folded; between 360–480 °C, crystal water was removed from AT; and above 500 °C, the structure water was removed. This caused the collapse of the channel, but the layer structure and morphology of AT remained unchanged [21]. Therefore, when AT was heated at 200 °C, it lost the amount of absorbed water and zeolite water, which increased the porosity; but as the channel was folded at the same time, which decreased the inner surface area, the sulfur absorption amount did not increase much. When AT was heated at 400 °C, it continued to lose crystal water, which loosened the channel, increased the specific surface area, and then improved the sulfur absorption amount. When the temperature reached 600 °C, AT lost most water and its crystal structure began to be destroyed, which relatively reduced the specific surface area. Thus, the sulfur absorption amount decreased. Therefore, the relative sulfur adsorption was higher at 400 °C compared to the other temperatures.

### 3.2. Characterization of AT @400 °C–S–PPy Composite

Figure 2 shows the FTIR curves of untreated AT, PPy, and the AT @400 °C–S–PPy composite. In the FTIR curve of untreated AT, the signal at 1651 cm^−1^ was attributed to the absorbed water, and the signals at 1026 and 985 cm^−1^ were respectively attributed to the stretching vibrations of the (Mg, Al)–Si–O bond and Si–O–Si (Al) bond [22]. In comparison, in the FTIR curve of the AT @400 °C–S–PPy composite, the signal of the absorbed water disappeared and the corresponding signals shifted to 1023 and 983 cm^−1^. In the FTIR curve of PPy, the characteristic signals appeared at 1538, 1461, and 1294 cm^−1^ and were separately attributed to the vibration of C=C in the pyrrole ring, C–C bond, and C–N bond [23]. In comparison, in the FTIR curve of the AT @400 °C–S–PPy composite, the characteristic peaks had blue-shifts to 1543, 1471, and 1307 cm^−1^. This was because when AT was coated with PPy, the electrons had transferred from the AT surface to PPy, leading to the increase in electron cloud density on PPy and the decrease in the electron cloud density on AT surface.

In Figure 3, the weight loss curve of the AT @400 °C–S–PPy composite is summarized and plotted against the curves of the purified AT, AT @200 °C–S, AT @400 °C–S, AT @600 °C–S, PPy, and elemental sulfur. In the curve of purified AT, the water departure started at 90–100 °C, which was due to the absorbed water from the air [18]. However, the AT @400 °C–S–PPy composite curve showed no obvious decomposition step at 90–100 °C. As sulfur was absorbed into the channels and pores of AT, little water was absorbed from the air [24]. In the curve of PPy, the decomposition step at 250–450 °C was attributed to the thermal decomposition of PPy, while in the curve of the AT @400 °C–S–PPy composite, the decomposition step at 200–350 °C was attributed to the removal of elemental sulfur, and the second decomposition step at 450 °C was attributed to the decomposition of PPy, which showed significantly higher thermal stability than PPy. The practical sulfur content of the AT @400 °C–S–PPy composite was about 50%, lower than the theoretical data, as some sulfur might have been lost during the preparation process.

The XRD spectra of the AT, AT @400 °C–S, and AT @400 °C–S–PPy composite can be seen in Figure 4. Diffraction peaks appear at 8.2°, 13.4°, 16.5°, 18.7°, 20.4°, 21.2°, 25.6°, 27.3°, 35.7°, and 42.2° in the curve for AT. In comparison, the intensity of the diffraction peak increases obviously in the curve for AT @400 °C–S due to the high content and good crystallinity of sulfur, which shows that sulfur was successfully absorbed in the AT. The diffraction peak for the AT @400 °C–S–PPy composite is consistent with that of AT @400 °C–S. It indicates that the coating of polypyrrole did not change the crystal structure of AT @400 °C–S and that polypyrrole existed on the surface of AT @400 °C–S in amorphous form [25].

To further demonstrate the structure of the AT @400 °C–S–PPy composite, SEM was utilized to investigate the morphologies of this composite. As depicted in Figure 5a, AT @400 °C had a uniform lamellar-like structure, and parallel channels were formed between tetrahedral bands. The structure of AT @400 °C–S (Figure 5(b1,b2)) was also lamellar-like, but there were sulfur parts on the surface. From Figure 5(c1,c2), we can see that PPy was evenly coated on the surface of the AT @400 °C–S, forming a stable lamellar structure. According to the International Union of Pure and Applied Chemistry (IUPAC), the definition of micropores, mesopores, and macropores is a relative conception. In the physical adsorption system, micropores are defined as pores with a diameter less than 2 nm; mesopores are pores with a diameter between 2 and 50 nm; and macropores are pores with a diameter more than 50 nm [26]. From the pore size distribution (Figure 6), it can be seen that the pore size distribution range of AT was wide, varying from 0.5 to 20 nm. The mesopores with a diameter between 2 and 7 nm were abundant, and there were fewer mesopores with a diameter between 7 and 20 nm. Micropores did not exist. After sulfur was loaded, the mesopores of AT @400 °C–S were focused in the range from 1.5 to 5 nm, and the pore volume decreased. This indicated that sulfur had occupied the channels and pores.

### 3.3. Electrochemical Measurements

The AT @400 °C–S–PPy composite was successfully employed as an active cathode material in the lithium–sulfur battery. Figure 7 depicts its cyclic voltammetry (CV) at a scan rate of 0.05 mV/s and a voltage range of 1.2–3.0 V. The peak potential difference can be calculated according to Equation (1) [27]:(1)$$\Delta E=\frac{2.3\mathrm{RT}}{\mathrm{nF}}$$
where ∆E: peak potential difference; R: Ideal Gas Constant, 8.314; T: absolute temperature; n: number of exchanged electrons; F: Faraday Constant.

During the first cycle of the scan, two distinct reduction peaks appeared at 2.3 and 2.0 V, which separately corresponded to the conversion of elemental sulfur S_8_ to the high-valence polysulfide S^2−^_x_ (x = 4–6) and the further reduction of high-valence polysulfide S^2−^_x_ to Li_2_S_2_, Li_2_S, etc. [28,29,30]. One oxidation peak appeared at 2.5 V, characteristic of the CV curve of elemental sulfur [31]. The CV curves of the second and the third cycle substantially coincided with each other, indicating that the cycle stability and reversibility of the battery were good [32].

The cycle performance curves and charge–discharge curves of the battery with the AT @400 °C–S–PPy composite as the cathode material exhibited two obvious plateaus (Figure 8a) at 2.3 and 2.0 V, characteristic of liquid electrolyte Li–S batteries [33,34]. According to the study by Rauh and Yamin [35,36], the plateaus at 2.3 and 2.0 V separately corresponded to the conversion of elemental sulfur S_8_ to S^2–^_x_ (x = 4–6) and the further reduction of S^2−^_x_ to Li_2_S_2_, Li_2_S, etc. In comparison, the charge–discharge curve of the contrasting PPy–S showed a sharper decrease after 10 cycles (Figure 8b). The calculation formula of battery capacity [27] is
(2)$$C=26.8n\frac{m}{F_{w}}$$
where C: theoretical specific capacity; m: the number of transferred electrons of each structural unit in the process of electrochemical reaction; F_W_: molecular weight of structural unit.

Therefore, the theoretical specific capacity of a lithium–sulfur battery is 1675 mAh/g. In the prepared system, the specific discharge capacity of this electrode was 1175 mAh/g in the first cycle. In the first 30 cycles, the specific discharge capacity gradually decreased to 696 mAh/g. After 100 cycles, the specific discharge capacity was maintained at 518 mAh/g, with a capacity retention rate of about 44%. From Figure 9, we can see that the initial specific capacity of the PPy–S electrode was 1209 mAh/g. After 100 cycles, the specific capacity gradually decreased to 420 mAh/g, with a capacity retention rate of about 35%. In comparison, the initial discharge specific capacity of the AT @400 °C–S–PPy composite electrode was 1175 mAh/g, and the capacity retention rate was about 44% after 100 cycles. The PPy–S electrode had a higher initial discharge specific capacity because the AT itself was not conductive and, after sulfur was adsorbed, the pores and channels of AT were blocked with sulfur and it could not absorb the polysulfides that were produced in the charge–discharge process. This resulted in the irreversible loss of active sulfur. At the same time, the insulated intermediates, like Li_2_S_2_ or Li_2_S, further blocked the pores and channels of AT, which hindered the active sulfur in AT from participating in the electrochemical reaction [37,38,39,40]. However, as the circulation continued, the active sulfur was redistributed, and the retained sulfur in the channels and pores was gradually activated. Therefore, the choked pores and channels were gradually cleaned, which enabled the subsequent absorption of the polysulfides that were produced during the charge–discharge process. This effectively inhibited the “shuttle effect” and improved the performance of the lithium–sulfur battery [41,42].

By gradually increasing the current density (0.1, 0.2, 0.5, 1, 2, 5 C) of the charge–discharge process (10 times cycled at each current density), the rate performance of the battery was obtained (Figure 10). At different current densities, the discharge specific capacities of the PPy–S electrode were about 1208, 638, 570, 483, 431, and 355 mAh/g, respectively. When the current density was readjusted to 0.1 C, the discharge specific capacity of the PPy–S electrode returned to 593 mAh/g. In comparison, the discharge specific capacities of the AT @400 °C–S–PPy composite electrode were about 1175, 790, 595, 448, 408, and 277 mAh/g. When the current density was readjusted to 0.1 C, the discharge specific capacity of the AT @400 °C–S–PPy composite electrode returned to 682 mAh/g. The result showed that at high current density, the pores and channels of AT were blocked, which also inhibited the transition of electrons. When the current density became lower—for instance, 0.1 C—the soluble polysulfide could be absorbed into the cleaned pores and channels of AT, so the AT @400 °C–S–PPy composite electrode could recover to higher specific discharge capacity. The comparison of the related work is shown in Appendix A.

## 4. Conclusions

In this experiment, we compared the sulfur absorption of untreated AT, purified AT, AT modified at 200 °C, AT modified at 400 °C, and AT modified at 600 °C. It was found that AT had the largest sulfur absorption amount after it was purified, acidified, and heated at 400 °C. Therefore, the AT @400 °C–S composite was chosen to be the template to prepare PPy, and a novel AT @400 °C–S–PPy composite with a lamellar nanostructure was successfully synthesized and then characterized by infrared spectroscopy (IR), TGA, and SEM. The AT @400 °C–S–PPy composite was further utilized as the cathode material for Li–S batteries. The first specific capacity of this kind of battery reached 1175 mAh/g at a 0.1 C current rate, and it remained at 518 mAh/g after 100 cycles with capacity retention close to 44%. In the rate test, compared to the PPy–S cathode material, the AT @400 °C–S–PPy cathode material showed a lower capacity at high current density, but it returned to a higher capacity when the current returned to a low current density. It was clearly confirmed that the pores and channels in AT can absorb the polysulfides produced during the charge–discharge process, effectively inhibiting the “shuttle effect” and improving the performance of the Li–S battery. From the comparison with related work (shown in the Appendix A), it is obvious that the lamellar AT @400 °C–S–PPy composite made full use of the natural structure of AT and shows great value for the material construction of Li–S batteries.

## Figures and Tables

**Figure 1 membranes-11-00483-f001:**
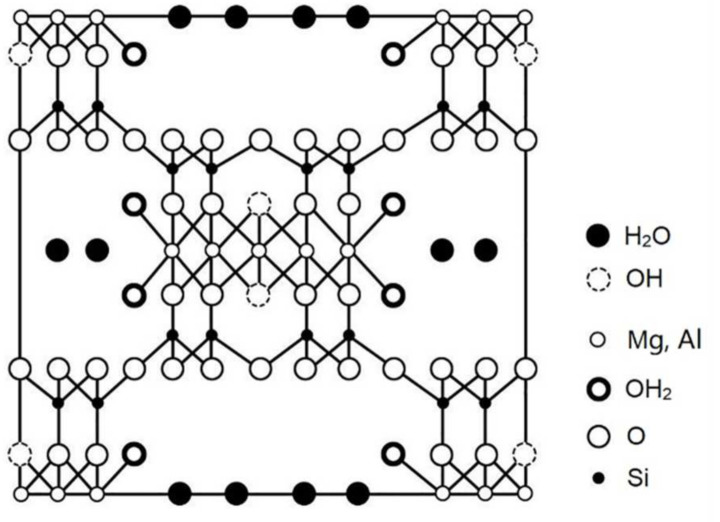
The molecular structure of attapulgite (AT) [14].

**Figure 2 membranes-11-00483-f002:**
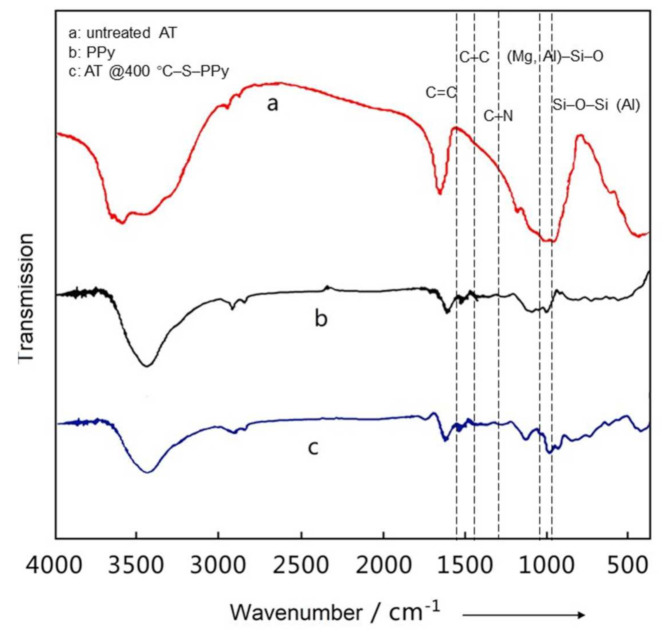
The FTIR curves of untreated AT, polypyrrole (PPy), and AT @400 °C–S–PPy composite.

**Figure 3 membranes-11-00483-f003:**
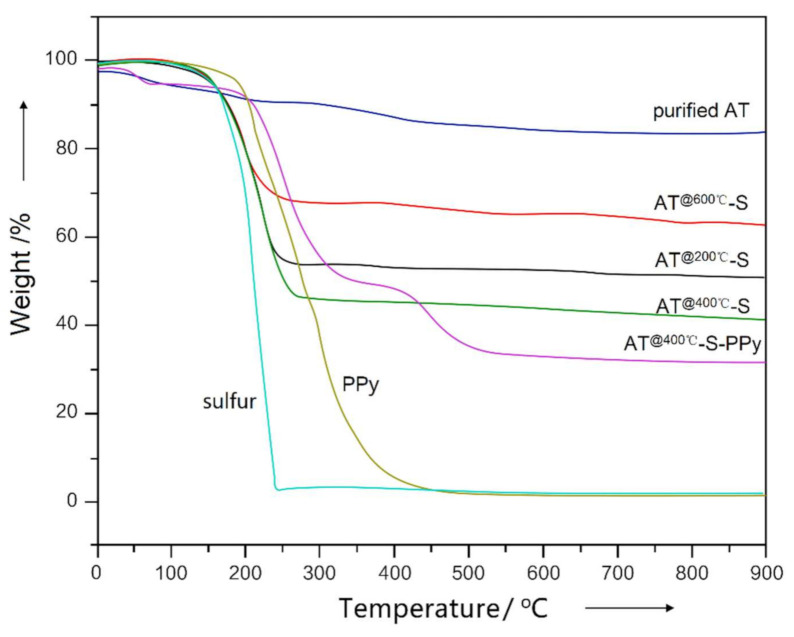
The TGA curves of purified AT, AT @200 °C–S, AT @400 °C–S, AT @600 °C–S, sulfur, PPy, and AT @400 °C–S–PPy composite.

**Figure 4 membranes-11-00483-f004:**
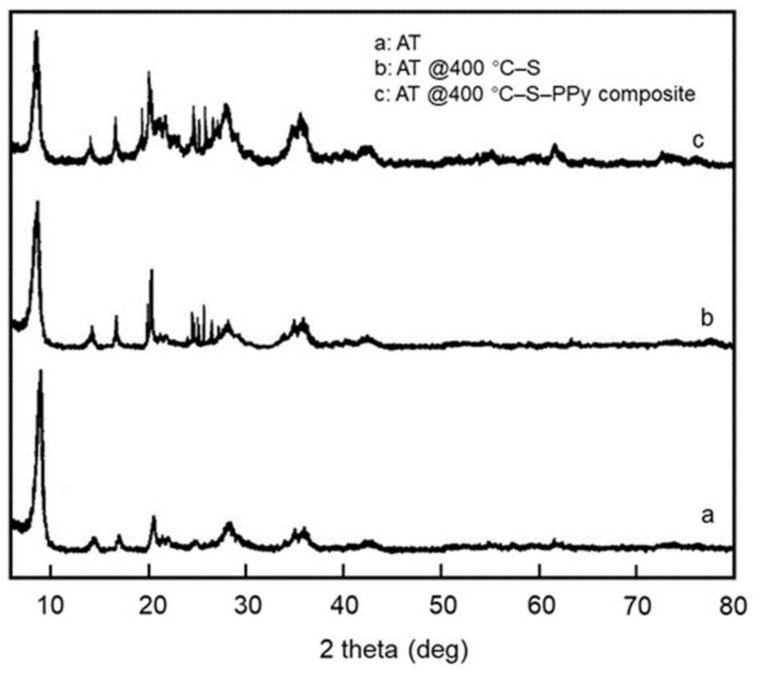
The XRD spectra of the AT, AT @400 °C–S, and AT @400 °C–S–PPy composite.

**Figure 5 membranes-11-00483-f005:**
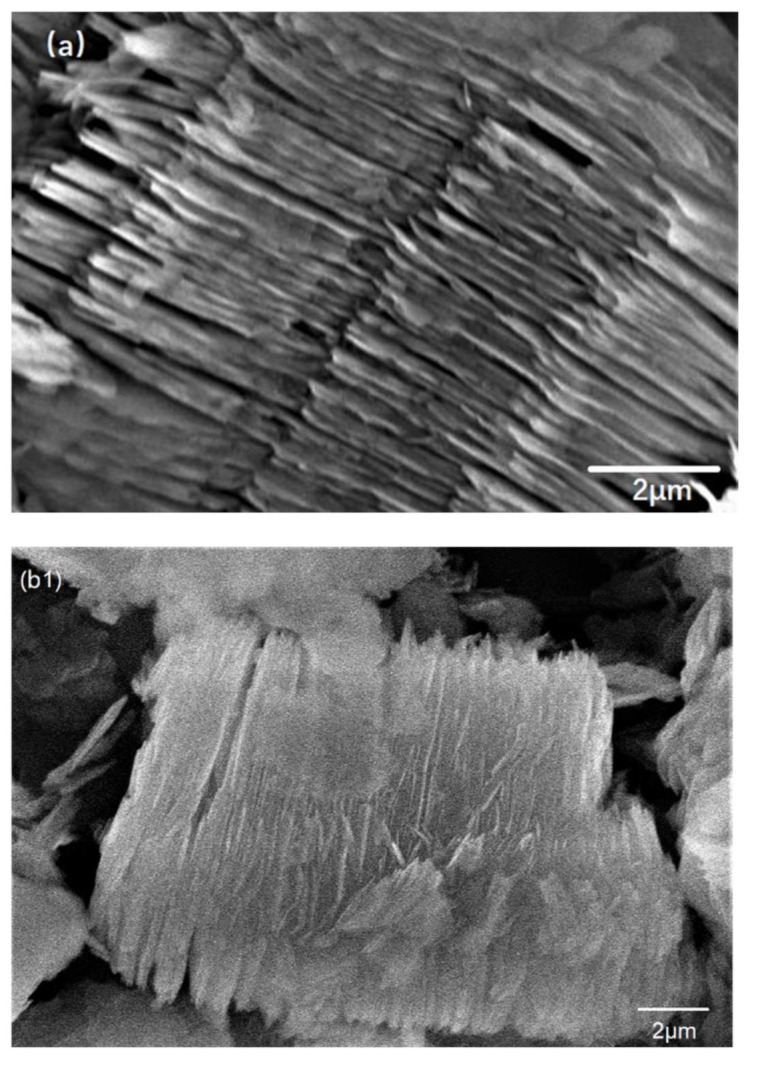
(**a**) Scanning electron microscope image of AT @400 °C; (**b1**) scanning electron microscope image (2 μm) of AT @400 °C–S; (**b2**) scanning electron microscope image (10 μm) of AT @400 °C–S; (**c1**) scanning electron microscope (2 μm) of AT @400 °C–S–PPy; (**c2**) scanning electron microscope (10 μm) of AT @400 °C–S–PPy.

**Figure 6 membranes-11-00483-f006:**
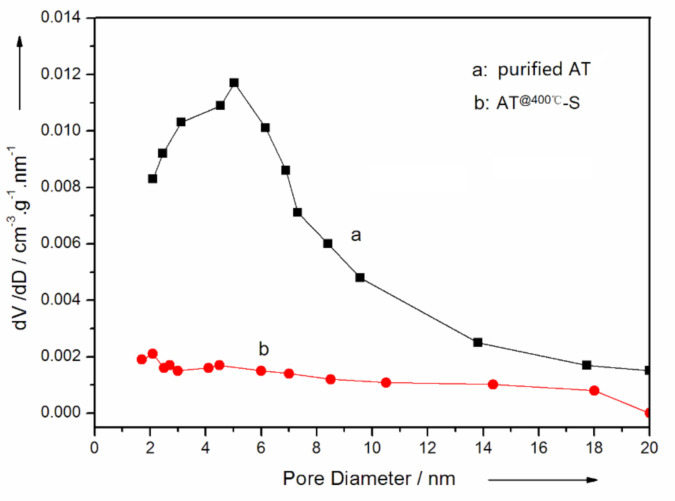
The pore size distribution curves of purified AT and AT @400 °C–S composite.

**Figure 7 membranes-11-00483-f007:**
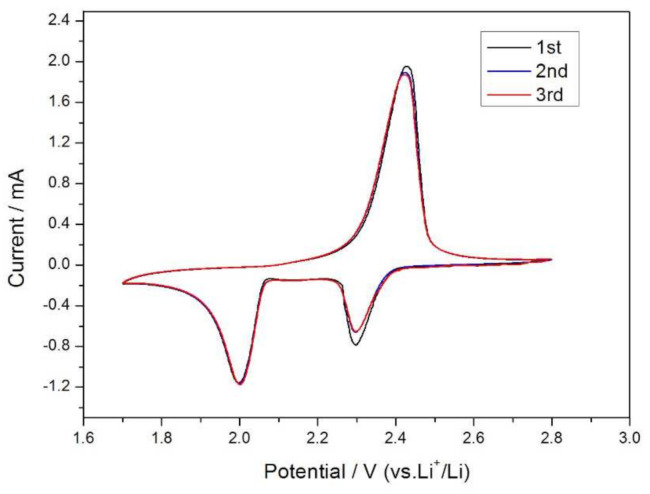
CV curves of the AT @400 °C–S–PPy composite electrode.

**Figure 8 membranes-11-00483-f008:**
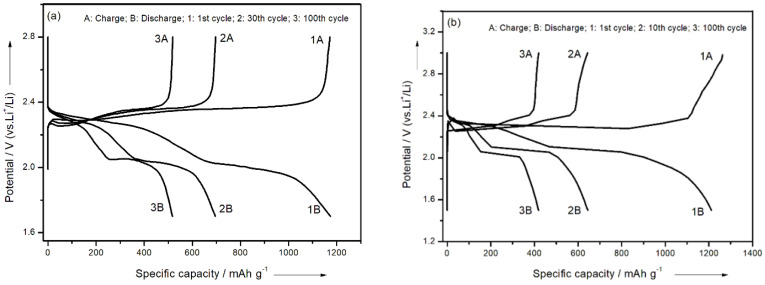
(**a**) The charge–discharge curves of the AT @400 °C–S–PPy composite electrode; (**b**) the charge–discharge curves of the PPy–S composite electrode.

**Figure 9 membranes-11-00483-f009:**
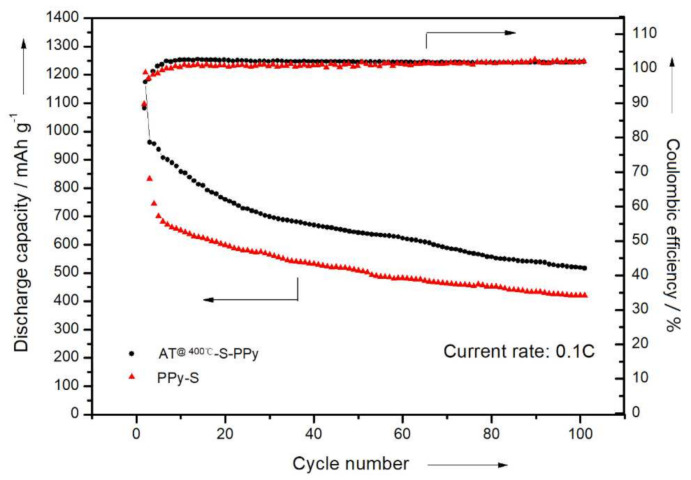
The cycle performance and Coulombic efficiency curves of the PPy–S electrode and AT @400 °C–S–PPy composite electrode.

**Figure 10 membranes-11-00483-f010:**
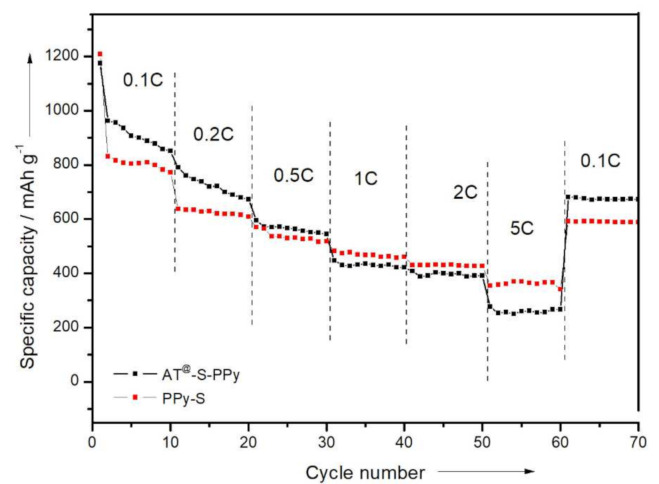
The rate performance curves of the PPy–S electrode and the AT @400 °C–S–PPy composite electrode.

**Table 1 membranes-11-00483-t001:** Relative sulfur absorption amount in different sulfur-containing AT systems.

AT System	Relative Sulfur Absorption Amount (wt.%)
AT1–S	21
AT2–S	45
AT @200 °C–S	47
AT @400 °C–S	58
AT @600 °C–S	28

## Data Availability

Not applicable.

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
