# Peer review of "Lamellar Polypyrene Based on Attapulgite–Sulfur Composite for Lithium–Sulfur Battery"

_membranes, 2021, doi:10.3390/membranes11070483_

Round 1
Reviewer 1 Report
After careful evaluation of the manuscript, I recommend for major revision before publication in Membranes journal. The reason for major revision is as follows.
Comments:
- The authors used PPY as the abbreviation for the polypyrene which is not correct as PPy refers to polypyrrole.
- Why the relative sulfur adsorption is higher at 400 C compared to the other temperature need to be discussed.
- The bands in the FT-IR need to be denoted in the revised manuscript.
- Why the rate capability of the device shows linear degradation in the specific capacity during initial cycles compared to the PPy-S device.
- Physicochemical characterization such as XRD, Raman can provide more insight regarding the prepared materials. Hence author need to provide in the revised manuscript.
Author Response
Dear reviewer,
Thanks for all youe suggestions and please see the attachment.
Best wishes!

Reviewer 2 Report
Review comments on Membranes,
Manuscript Number: membranes-1257870
Manuscript title: Lamellar Polypyrene Based on Attapulgite–Sulfur Composite 2 for Lithium-Sulfur Battery
The novelty of the work is very good. The present manuscript should be accepted after a major revisions following the comments:
- If possible, need to add the XRD data of the composite systems.
- Both the equations of calculation of CV and CD are need to show with proper reference.
- The manuscript needs to check the English editing for improvement of English.
- Use the proper refernces related this work such as https://doi.org/10.1039/C6RA18811G, https://doi.org/10.1016/j.jiec.2018.01.023
Author Response
Dear reviewer,
Thanks for all youe suggestion and please see the attachment.
Best wishes!
